# Impact of dominance rank specification in dyadic interaction models

Alexander Mielke [iD] [1,2,3] *

**1** School of Psychology and Neuroscience, University of St Andrews, St Andrews, United Kingdom,
**2** Primate Models for Behavioural Evolution Lab, School of Anthropology and Museum Ethnography, Oxford, United Kingdom, **3** Taï Chimpanzee Project, Centre Suisse de Recherches Scientifiques en Côte d'Ivoire, Abidjan, Côte d'Ivoire

\* mielke.alexand@gmail.com

## Abstract

Dominance rank is a vital descriptor of social dynamics in animal societies and regularly used in studies to explain observed interaction patterns. However, researchers can choose between different indices and standardizations, and can specify dyadic rank relations differently when studying interaction distributions. These researcher degrees of freedom potentially introduce biases into studies and reduce replicability. Here, I demonstrate the impact of researcher choices by comparing the performance of different combinations of rank index, standardization, and model specification when explaining dyadic interaction patterns in sooty mangabeys (*Cercocebus atys atys*). I show that while no combination consistently performed best across interaction types (aggression, grooming, proximity, supplants), model specifications allowing for nonlinear patterns performed better than other models on average. Choices made in pre-processing and model building impacted model performance and subsequent interpretation of results. Researchers could end up describing social systems differently based on the same data. These results highlight the impact of researcher choices in the processing of behavioural data and potential limitations when using indirect species comparisons in animal behaviour research. To increase repeatability, researchers could make the impact of their processing choices more transparent and report results using a variety of indices and model specifications.

## Introduction

Dominance hierarchies are one of the most conspicuous structural features of animal societies [1]. Calculating dominance hierarchies accurately, determining their linearity and temporal structure, and understanding how they influence behaviour and fitness has thus long been central to the study of animal behaviour [2]. Dominance rank, as a measure of the way competitive advantage is distributed within a group, is used frequently as a predictor or outcome variable in models describing why animals act the way they do [3]. Different indices for calculating dominance hierarchies are available to researchers, with different assumptions about the nature of dominance and the underlying interaction distribution [4–11]. For example, indices can assume linearity of hierarchies or not [7], account for temporal change or not [4], and can

**Data Availability Statement:** Scripts and data are available from here: https://github.com/AlexMielke1988/Mielke-Mangabey-Ranks.

**Funding:** Wenner-Gren Foundation Dissertation Fieldwork Grant, Grant Number 9095, http://www.wennergren.org/ British Academy Newton

International Fellowship, NIFBA19\191052, https://www.thebritishacademy.ac.uk/ Leverhulme Trust Early Career Fellowship, ECF-2021-642, https://www.leverhulme.ac.uk/ The funders had no role in study design, data collection and analysis, decision to publish, or preparation of the manuscript.

**Competing interests:** The authors have declared that no competing interests exist.

weigh interaction intensity differently [12]. Once an index has been calculated, it can be standardised to make it more meaningful for the system under consideration, for example by creating an ordinal rank hierarchy or accounting for group size [13]. Researchers want to make the correct choices when determining which values to include in their analyses; however, the large variety of solutions in circulation indicates that no optimal solution for all species and circumstances exists. The existing diversity of researcher choices poses risks to the replicability of studies [14].

The uncertainty for researchers increases when studying the impact of dominance rank on interactions between individuals. Here, not only the individual rank is of interest; we often want to represent how the dominance ranks of both sender and receiver determine interaction patterns. Importantly, rank can influence partner choice in diverse ways: individuals can choose a partner because of the partner's rank, the simple fact that they are higher-ranking than the partner, the distance between ranks, or a nonlinear interaction between their own rank and that of the partner. For example, in tufted capuchin females, high-ranking individuals gain more grooming (effect of dominance rank), individuals groom similarly-ranked individuals more than expected (effect of rank distance), but high- and low-ranking individuals differ in how freely they can choose partners (nonlinear effect of groomer rank; [15]. These relationships can be captured using different specifications in statistical models: for example, we can include a fixed effect for each individual rank in a linear model, or the rank difference between them. But just as each dominance index makes assumptions about the underlying distribution of rank [7], any decision about how individual and dyadic ranks are included in models makes assumptions about power dynamics in the group. When fitting statistical models, we decide on one way to represent power relations to describe variation in our outcome variable as accurately as possible. However, model specifications might lead researchers to different conclusions about their study system. For example, for the above example of tufted capuchin females [15], if a researcher models only the dominance ranks of the two groomers, they might find that high-ranking individuals received more grooming. With the same data, another researcher who included only the rank difference would interpret the results to indicate that individuals were attracted to similarly-ranked group members. In isolation, these would point to fundamentally different processes underlying grooming partner choice. We do not currently know the potential impact of pre-processing and analytical choices on interpretations of dyadic interaction patterns. In this study, I apply diverse ways to define dominance rank in statistical models to the same dataset of sooty mangabey (*Cercocebus atys atys*) interaction patterns to explore how differences in pre-processing and model specifications influence conclusions about the social system.

Researchers face two competing responsibilities to increase the credibility of their work: on one hand, they should describe the dominance structure of their study system as accurately as possible; on the other hand, their results should be replicable and comparable with the existing literature [16]. Given the variety of choices researchers can make in this context, it is easy to see that replicability between studies can suffer [17]. Currently, researcher choices are mainly hidden–readers are presented with one combination for dominance index, standardisation, and model specification. Different iterations that might have been tried in pre-processing but were not chosen might not be transparently reported [18], inflating the reported expected number of incorrect rejections in a frequentist framework [19, 20]. If results were conditional on researcher choices, comparing independent studies becomes strenuous. For example, when studying infant handling in primates, ranks can be described using the rank difference between two individuals [21], 'higher/lower-ranking than mother' categories [22], or absolute rank distance [23]. While the authors of each individual study chose their methodology with a specific hypothesis in mind, we do not know whether differences between studies show species

differences or reflect analytical choices. Certain model specifications might also limit possible interpretations: for example, using absolute rank distance assumes symmetrical effects and precludes different effects for high- and low-ranking group members. A researcher designing a new study on the same subject would have to decide which approach to replicate, and why.

While presenting the impact of different choices in this article, I argue for transparent and open reporting of different analytical pathways using a 'multiverse analysis' approach for future studies [14, 24]. Rather than choosing one rank index/standardization/model specification combination (e.g., proportional Elo index entered as main effects), researchers could consider calculating and reporting all possible combinations, to show that their interpretation is not conditional on the choices they made [19]. This transparent approach has been highlighted as a viable strategy to handle the uncertainty arising from researcher degrees of freedom in the scientific process when multiple analytical choices are available [14]: as uncertainty cannot be removed, an honest approach can counter selective reporting and increase comparability across studies. As data pre-processing and analysis pipelines become more accessible with open data and scripts, including multiple results to rule out conditionality of interpretations on researcher degrees of freedom becomes increasingly feasible.

In this study, I compare the impact of different processing steps on the analysis of dyadic interaction rates, by varying dominance indices, index standardisation, and model specification. Optimally, these factors would have little influence, and choices along the data analysis pipeline would minimally affect results and interpretations [18]. However, the hypothesis for this article is that different model specifications considerably influence model fit and interpretation across different social interaction types (aggressions, spatial proximity, feeding supplants, grooming). Recent studies [5, 7, 8, 11, 12, 25, 26] have compared and/or improved several dominance rank indices. Here, I focus on two commonly used dominance indices–David's Score [6] and (optimised) Elo rating [27] to test whether this choice influences results. I compared different standardizations—raw David's Scores or Elo rating values, proportional ranks (all ranks equidistant between 0 and 1, with 1 for the highest-ranked individual), and ordinal ranks (1 for highest-ranking individual, 2 for second highest etc)—because this choice has recently been shown to influence model outcomes in primate studies [13, 28], but insufficient information still exists on whether and how they influence models that represent interaction rates between dyads of individuals within groups. My focus lies on how the power relations between the individuals in a dyad are represented when fitting models (whether as main effects, rank distance variables, or interaction terms; see Table 1), given that no systematic information on the impact of this choice is currently available. My prediction is that

**Table 1. Different model specifications for rank in dyadic interaction models tested in this study.** + indicates that main effects were entered independently in the model, * indicates a statistical interaction term.

| Model Name | Model Terms | Information about |
|---|---|---|
| *Main Effects* | Rank Sender + Rank Receiver | Rank Position Sender, Rank Position Receiver |
| *Factor Higher-ranking* | Rank Sender + Higher-ranking yes/no | Rank Position Sender, Does the Sender outrank the Receiver? |
| *Rank Difference* | Rank Sender + (Rank Sender– Rank Receiver) | Rank Position Sender, Rank Position Receiver, Distance in Rank |
| *Absolute Rank Difference* | Rank Sender * Abs(Rank Sender– Rank Receiver) | Rank Position Sender, Distance in Rank |
| *Interaction* | Rank Sender * Rank Receiver | Rank Position Sender, Rank Position Receiver, Linear Connection between them |
| *Nonlinear Interaction* | Tensor Product (Rank Sender, Rank Receiver | Rank Position Sender, Rank Position Receiver, Nonlinear connection between them |

different model specifications of rank differences vary in their prediction error of real-world primate data and lead to different interpretations of the impact of dominance rank on social interactions.

## Methods

### Ethics statement

Permissions to conduct the research were granted by the Ministries of Research and Environment of Ivory Coast (379/MESRS/GGRSIT/tm) and Office Ivorien des Parcs et Reserves. Methods were approved by the Ethikrat der Max-Planck-Gesellschaft (4.08.2014).

### Data set

I used data collected on adult female sooty mangabey social interactions in Taï Forest National Park, Cote d'Ivoire, by trained field assistants and me between January 2014 and June 2017 as part of the ongoing long-term data collection for the Taï Chimpanzee Project [29–32]. I selected three 1-year blocks of data on aggression (n = 1033 events), feeding supplants (n = 818 events), grooming (n = 3252 events), and spatial proximity within 1m (n = 6080 events) for all adult females (above 4.5 years) present in the group for the entirety of each block (total: 25 females; range: 17–18 females per block). I only use adult females, as their hierarchies have been found to be clearly linear in a neighbouring community (h = 0.71; [33]) and this community (only 0.6% of supplants direct against the hierarchy; [34]) and they interact at relatively high rates, while the connection between male and female hierarchies is less clear and males interact at low rates in this group [30].

 The response variables were, respectively, counts of aggression events (defined as any instance of A threatening or physically attacking B), counts of feeding supplant events (defined as A displacing B from a food source, without overt threat), counts of proximity events (defined as B being the nearest neighbour of A within 1m when A sits down to eat or rest), and minutes of grooming directed from A to B. I did not differentiate between different intensities of aggression. For each year, the dataset contained each sender-receiver combination and the number of interactions observed between them in that period–i.e., each sender-receiver combination is present in the dataset up to three times (once per year), and each individual is included as sender and receiver for each year with each possible partner. Proximity, in contrast to the other interaction types, was symmetrical (i.e., A to B equals B to A), as no 'sender' and 'receiver' could be defined.

 We previously showed that the aggression and supplant data are internally consistent and therefore probably have low measurement error [34]. For the grooming and proximity distributions internal consistency is lower, because of low partner selectivity, the considerable number of dyads, and the relatively random distribution of proximity in this group of mangabeys [32]. For both grooming [31, 33, 35] and proximity [32, 33], dominance rank has been shown to affect sooty mangabey behaviour. All response variables are described in detail below.

### Dominance indices

I calculated dominance rank using two different metrics: normalized David's Score [6, 36] and optimized Elo ratings [27]. I chose the latter over Elo ratings without optimized k and start values as we did not observe any active rank changes between females over the course of this study [30]; thus, all changes in an individual's rank are due to demographic processes (individuals dying or coming of age). The optimised k value for the Elo ratings was therefore confirmed and calculated as 0 for all year blocks. The Elo value at the first day of the year was

chosen to represent the dominance hierarchy. Dominance hierarchies were computed using directional feeding displacements ('supplants'), which are highly linear in this group, with only 6 out of 818 supplants (or 0.7%) going against the established hierarchy. Normalized David's Scores were calculated using the 'steepness' package in R [37]. Elo ratings were calculated using the 'Model 3' script provided by [27]. Both indices assume linearity of hierarchy, which was given in the female mangabeys. For both normalized David's Scores and Elo ratings, I applied different standardisations: the raw values, the ordinal hierarchy based on the raw values (highest-ranking individual has rank 1, second highest has rank 2 etc.), and the proportional rank standardised between 0 and 1, with 1 being the highest-ranking individual [13]. Ordinal and proportional ranks assume equidistance of ranks.

## Model specifications

Here, I focus on six ways of representing two rank terms in statistical models of dyadic interactions, all of which have been used for animal behaviour research (Table 1). I only used numerical representations of dominance rank–categorical representations (e.g., 'high/medium/low') are still in use [38] but reduce information and add even further researcher degrees of freedom (e.g., about the number of categories and cut-offs). All models include the sender rank as a main effect. _Main Effects model_: I included both ranks as main effects, which informs us about the variation in partner choice due to whether the sender is high or low in rank and the receiver is high or low in rank but does not represent the relation between the two. _Factor Higher-ranking_: I fitted a model using a factorial term defined by whether the sender was higher- or lower-ranking than the receiver (based on the raw Elo index), which captures one simple aspect of their relationship but omits the rank distance and the actual receiver rank. _Rank Difference_: I fitted a model with a rank difference term (subtracting the receiver rank from the sender rank), which is positive if the sender outranks the receiver but also captures information on the distance between them. _Absolute Rank Difference_: The rank difference term cannot be fitted in interaction with the sender rank because the two are tightly bound to each other–how much higher or lower in rank an individual can be overall is determined by the sender rank. In contrast, the absolute rank distance term is small when individuals are close in rank, and large if they are far apart, but omits information about whether the receiver outranks the sender. _Interaction_: I fitted one model representing the interaction term between the sender and receiver ranks. The statistical interaction between the two rank variables can represent both the direct impact of sender and receiver rank but can also account for differences in partner choice between high—and low-ranking senders, as long as these are linear: for example, high-ranking senders can supplant all group members, while low-ranking senders can supplant only low-ranking group members. _Nonlinear Interaction_: Lastly, I fitted a model including the tensor product smooths [39] of the two rank variables as a nonlinear representation of their interaction, using the 'mgcv' package [40]. Like splines [41], tensor product smooths express the nonlinear regression curve of parameters as localised pieces that maximise smoothness–with the difference that splines represent one variable, while the tensor product smooths used here represent the interaction between two variables. This approach is preferable to using polynomial terms for one of the parameters [31] because tensor product smooths allow for better extrapolation and give locally more accurate predictions [39]. However, the model is considerably more complex and can tend towards overfitting [41] This model can capture the main effects of both ranks, and identify linear interaction effects, but should also be able to represent nonlinear partner choice, for example if both high- and low-ranking group members preferably groom individuals who are close in rank. In the following, I use all

the model terms defined in Table 1 to describe the distribution of social interactions in sooty mangabeys to determine which of these captures most of the variation in observed patterns.

## Models

I tested how different scores (modified Elo ratings, David Scores), standardisation (raw, standardised 0–1, ordinal ranks) and model specifications (main effects, factor higher/lower ranking, rank difference, absolute rank difference, interaction term, nonlinear interaction term) influence distribution of interactions in sooty mangabeys. I did not control for any other variables (e.g., age). All rank variables were z-standardised before being entered into the models [42]. In all models, offset terms were used to account for the dyadic observation effort (as individuals could be both sender or receiver of interactions when either was followed). All models were fitted using Bayesian generalized linear mixed models in the 'brms' package [43] in R v4.1.2 [44]. For the aggression and supplant model, I used zero-inflated Poisson error structure, as a large number of dyads had no recorded interactions for each type. For proximity, I fitted the model using Poisson error structure. For grooming, the most appropriate error structure was determined to be zero-inflated negative binomial. All models contained the log-transformed observation effort in hours as offset term.

Models were fitted using 3000 iterations on three chains. For all fixed effects in all models, I used weakly informative, normally distributed priors [45] with a mean of 0 and a standard deviation of 1. All models contained the year, sender identity, receiver identity, and dyad identity as random effects. Model performance was evaluated and compared using the leave-one out cross validation information criterion as available in the 'loo' package [46], as a measure of prediction error of the model posterior distribution. My assumption here is that, given that the outcome remains constant, the model with the smallest LOO-IC predicts the outcome most accurately, and that models with lower leave-one out cross validation information criterion and higher Bayesian R2 [47] can be considered to be 'better' than models with higher LOO-IC or lower Bayesian R2. All models within a social interaction type were compared against each other, and I report the difference between the LOO-IC of each model with the best model in the set (deltaLOO); the best model therefore has a deltaLOO of 0. I report Bayesian R2 values for the best model to give a sense of explained variance [47]. I also report the interpretation researchers would have arrived at using the fixed effect results of each of the models, to demonstrate that these choices matter for the publication process. The Supplementary Material contains the graphs for impact of the two dominance ranks in the nonlinear models with proportional Elo-based ranks for all four interaction types, to help readers visualise the described effects. Effects were interpreted as meaningful if the 95% credible interval of the posterior distribution did not include 0, and plots were used to establish the direction of effects.

## Results

### Aggression

Comparing all models for the distribution of aggression interactions in the group, the simple factor describing whether the sender was higher- or lower-ranking than the receiver had the highest out-of-sample prediction accuracy (Fig 1). This is explained by the distribution of zeros–lower-ranking individuals almost never attack higher-ranking individuals in this group. The explained variance of the best model for aggression was $R^2 = 0.53$. Of the other models, everything else being equal, the Elo-based models on average outperformed the David's Score based models (Fig 1A). The proportional and ordinal standardizations did not show consistent differences, but both outperformed the raw values (Fig 1B). All these differences were minor compared to the impact of the model specification (Fig 1C). Models that can represent

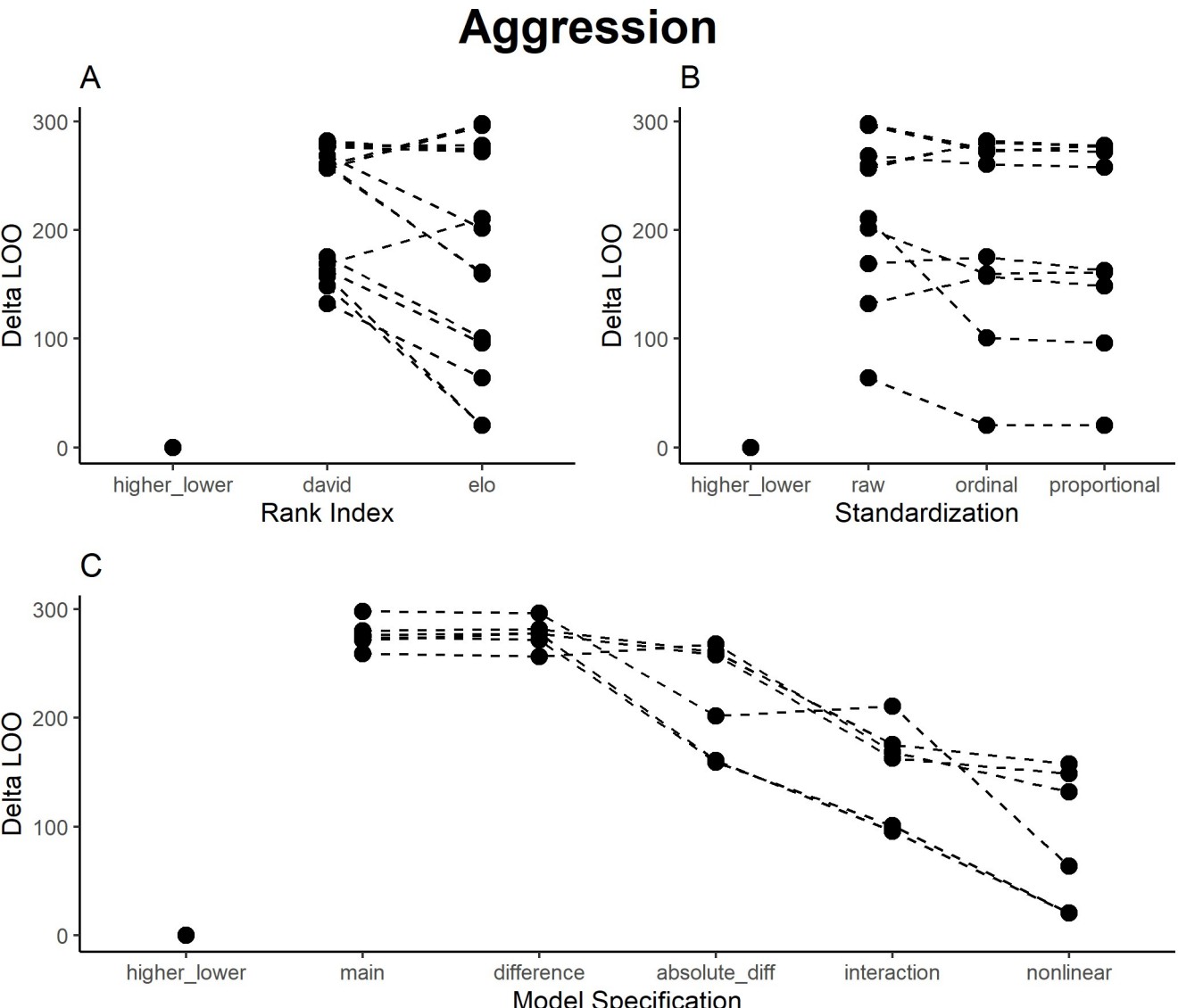

**Fig 1. Results of model comparison for aggression interactions.** The y-axis portrays delta LOO-ICs (distance to best model)—the best model is therefore set to 0, and higher scores indicate poorer performance. Points indicate models (split by the respective category), while lines connect otherwise comparable models (e.g., the main effects ordinal scale models for both Elo ratings and David's Scores).

nonlinear rank distances, especially the nonlinear interaction and interaction models, but also the model including absolute rank distance, performed considerably better than those using only main effects or the rank difference. This would considerably affect interpretations of results (Table 2): Most well-performing models found that high-ranking senders showed more aggression, and aggression was directed down the hierarchy, but preferentially against closely ranked group members (S1 Fig in S1 File). The model using only main effects, modelling only whether either individual was high- or low-ranking, failed to capture these dynamics. The higher/lower rank factor and the rank difference models would have captured the down-the-hierarchy aspect, implying that all individuals indiscriminately attack lower-ranking group members. The absolute rank distance would have captured the increased aggression towards closely ranked group members but would indicate that this effect was distributed evenly for

**Table 2. Interpretation of Elo results for aggression.** Model interpretations are displayed for all models using Elo ratings only to facilitate interpretation. Downwards arrows indicate that lower-ranking individuals show higher rates, upward arrows indicates that higher-ranking individuals show higher rates. Interactions can show Down The Hierarchy (DTH; Targeting lower-ranking individuals), Closely Ranked Receiver (CRR; Targeting individuals with similar rank); Up The Hierarchy (UTH; Targeting higher-ranking individuals), or a mix of those.

| Model | Standardization | DeltaLOO | Sender Rank | Receiver Rank | Interaction |
|---|---|---|---|---|---|
| *Factor Higher-ranking* | - | 0 | ↓ | - | DTH |
| *Main Effects* | Raw | 297.9 | ↑ | ↓ | - |
| | Ordinal | 274.3 | ↑ | ↓ | - |
| | Proportional | 271.8 | ↑ | ↓ | - |
| *Rank Difference* | Raw | 296.4 | - | - | DTH |
| | Ordinal | 271.7 | - | - | DTH |
| | Proportional | 277.9 | - | - | DTH |
| *Absolute Rank Difference* | Raw | 201.6 | ↑ | - | CRR (especially high-ranking sender) |
| | Ordinal | 159.4 | ↑ | - | CRR (especially high-ranking sender) |
| | Proportional | 160.9 | ↑ | - | CRR (especially high-ranking sender) |
| *Interaction* | Raw | 210.7 | ↑ | ↓ | High-ranking: CRR and DTH; Low-ranking: CRR |
| | Ordinal | 100.9 | ↑ | ↓ | High-ranking: CRR and DTH; Low-ranking: CRR |
| | Proportional | 96.0 | ↑ | ↓ | High-ranking: CRR and DTH; Low-ranking: CRR |
| *Nonlinear Interaction* | Raw | 63.9 | ↑ | - | High-ranking: CRR and DTH; Low-ranking: CRR |
| | Ordinal | 20.4 | ↑ | - | High-ranking: CRR and DTH; Low-ranking: CRR |
| | Proportional | 20.5 | ↑ | - | High-ranking: CRR and DTH; Low-ranking: CRR |

high- and low-ranking senders. The interaction and nonlinear interaction models allowed for more complex interpretations and different patterns at different points in the hierarchy, but interpretations were also more ambiguous based on plots alone. Here, it seemed that low-ranking individuals receive more aggression because they are victims both of other low-ranking and high-ranking individuals, while low-ranking never attacked high-ranking individuals.

## Grooming

For grooming, models using David's Scores generally performed better than those using Elo ratings (Fig 2A), and again raw values performed worse than both ordinal and proportional values (Fig 2B)–however, overall differences between model performances were less pronounced than in the aggression models. The best models (with small differences between them) were those that could quantify closeness in rank (Fig 2C): the nonlinear interaction terms for the David's Scores (both proportional and ordinal) as well as the absolute rank difference models. The explained variance of the best model for grooming was $R^2 = 0.31$. Main effects models, the higher/lower factor, and the rank distance consistently performed worse than other models. Again, using different models would have led to different interpretations. The consensus was that there was a tendency to groom individuals who are close in rank, and that this is particularly pronounced in the highest-ranking individuals (Table 3, S2 Fig in S1 File). In some models, higher-ranking individuals tended to receive more grooming, and lower-ranking tended to give more grooming, however, this effect was not consistent and not necessarily linear (with medium-ranked individuals often performing the same as high-ranking individuals). Some of the main effects, rank distance, and the higher/lower factor model showed that high-ranking individuals received more grooming, indicating that grooming went up the hierarchy. The other models (absolute rank difference, interaction and nonlinear interaction models) indicated grooming of those close in rank, with the nonlinear interaction model also indicating that this effect was stronger in medium- and high-ranking individuals.

# Grooming

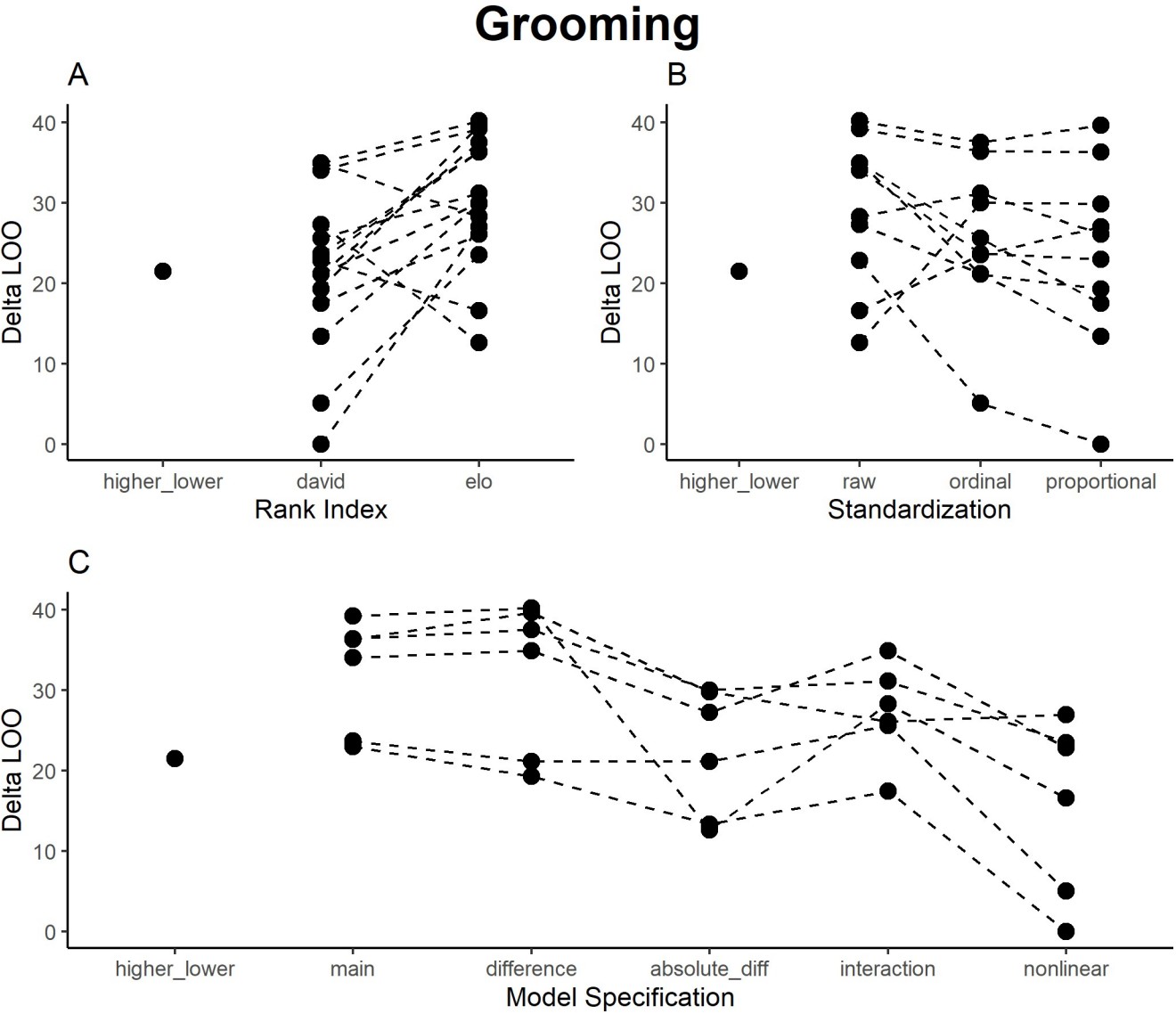

**Fig 2. Results of model comparison for grooming interactions.** The y-axis portrays delta LOO-ICs (distance to best model)—the best model is therefore set to 0, and higher scores indicate poorer performance. Points indicate models (split by the respective category), while lines connect otherwise comparable models (e.g., the main effects ordinal scale models for both Elo ratings and David's Scores).

## Proximity

The proximity data were symmetrical (A → B = = B →A), allowing us to test how models cope with this additional oddity in the data. For proximity, models using David's Scores and Elo ratings performed roughly similar (Fig 3A), and again raw values performed worse than ordinal values, which in turned performed worse than proportional values (Fig 3B). In fact, across models, raw values detected different patterns than the two standardizations (Table 4), and anyone basing their interpretation on those results would describe a different social system. In raw values, the sender and receiver main effects both had a strong positive effect (high-ranking individuals generally show higher levels of proximity), while for both standardizations, the two main effects had strong negative effects. The raw value models indicated a preference for closely ranked partners for proximity across sender ranks, while the other models indicated a

**Table 3. Interpretation of Elo results for grooming.** Model interpretations are displayed for all models using Elo ratings (similar results for David's Scores). Downward arrows indicates that lower-ranking individuals show higher rates, upward arrows indicates that higher-ranking individuals show higher rates. Interactions can show Down The Hierarchy (DTH; Targeting lower-ranking individuals), Closely Ranked Receiver (CRR; Targeting individuals with similar rank); Up The Hierarchy (UTH; Targeting higher-ranking individuals), or a mix of those.

| Model | Standardization | DeltaLOO | Sender Rank | Receiver Rank | Interaction |
|---|---|---|---|---|---|
| *Factor Higher-ranking* | - | 21.5 | - | - | UTH |
| *Main Effects* | Raw | 39.2 | - | ↑ | - |
| | Ordinal | 36.4 | - | - | - |
| | Proportional | 36.3 | - | - | - |
| *Rank Difference* | Raw | 40.2 | ↑ | - | UTH |
| | Ordinal | 37.5 | - | - | - |
| | Proportional | 39.6 | - | - | - |
| *Absolute Rank Difference* | Raw | 12.6 | - | - | CRR |
| | Ordinal | 30.0 | - | - | CRR |
| | Proportional | 29.8 | - | - | CRR |
| *Interaction* | Raw | 28.3 | - | ↑ | CRR |
| | Ordinal | 31.2 | - | - | CRR |
| | Proportional | 26.1 | - | - | CRR |
| *Nonlinear Interaction* | Raw | 16.6 | - | ↑ | High-ranking: CRR; Low-ranking: CRR and UTH |
| | Ordinal | 23.5 | ↓ | ↑ | High-ranking: CRR; Low-ranking: CRR and UTH |
| | Proportional | 27.0 | - | - | High-ranking: CRR; Low-ranking: CRR and UTH |

preference for closely ranked partners in low- and medium-ranking individuals, while high-ranking individuals showed no clear preference (S3 Fig in S1 File). This might be because I removed males from the analysis. The different result for raw values is concerning, given the raw values were highly correlated ($> 0.9$) with both standardizations across indices. Given the inferior performance of the raw values in the model comparison, these models likely fail to predict the actual group patterns, but this would not be apparent to a researcher focusing exclusively on these values. The best model to detect proximity was the nonlinear interaction model for the proportional Elo ratings (explained variance: $R^2 = 0.73$) (Fig 3C). Despite the symmetry of the data, the higher/lower factor and rank difference would have indicated directional effects. The absolute rank distance, interaction, and higher/lower factor models performed worse than the other models.

## Supplants

For supplants, models using Elo ratings generally performed better than David's Scores (Fig 4A), and raw values performed worse than both ordinal and proportional values (Fig 4B). However, as in aggression, the model with a simple factor denoting that the sender was higher-ranking performed best (explained variance: $R^2 = 0.59$)–not surprisingly, given that supplants are used to make the dominance hierarchy. Among the other models, those that encode that supplants generally go down the hierarchy and differences in patterns across the dominance hierarchy–interactions and nonlinear interactions–fared better than the absolute difference, rank difference, and main effects models (Fig 4C). The best models (apart from the higher/lower factor model) were the nonlinear interaction models of proportional and ordinal David's Scores and Elo ratings. Results closely resembled the aggression models: Main effects, the higher/lower factor, and the rank difference would have indicated supplants going down the hierarchy, the absolute rank difference would have indicated supplants to those close in rank, and the interaction and nonlinear interaction models captured a mix of the two effects (Table 5, S4 Fig in S1 File).

# Proximity

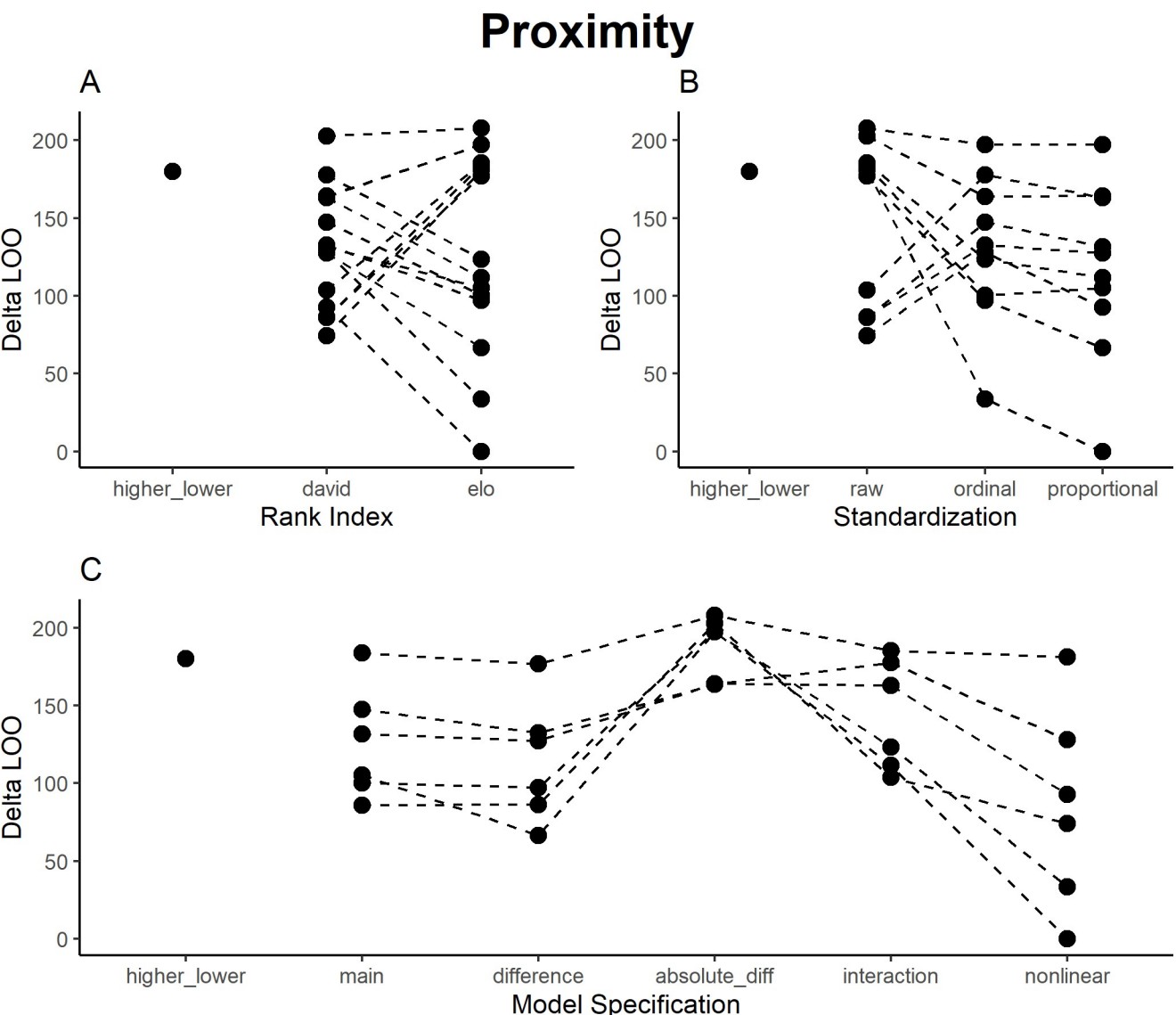

**Fig 3. Results of model comparison for proximity interactions.** The y-axis portrays delta LOO-ICs (distance to best model)–the best model is therefore set to 0, and higher scores indicate poorer performance. Points indicate models (split by the respective category), while lines connect otherwise comparable models (e.g., the main effects ordinal scale models for both Elo ratings and David's Scores).

## Discussion

My results showed that the choice of dominance rank index, standardization, and model specification matter when interpreting how ranks affect dyadic interaction patterns in primate societies. This is true for all models using dominance rank in some form–it becomes even more central for dyadic models, which have to capture the interactional nature of animal social life. For all interaction types, the results showed that different researchers, using the same data, would have reached different interpretations of the social life of sooty mangabeys. Only a handful of studies exist for most animal species, comparisons between species are usually indirect, and results across studies are aggregated to make higher-level statements about evolutionary forces underlying sociality (e.g., socio-ecological models, [48]. Without sufficient control, there might be a risk of creating non-replicable results that come to represent what we know

**Table 4. Interpretation of Elo results for proximity.** Model interpretations are displayed for all models using Elo ratings (similar results for David's Scores). Downward arrows indicate that lower-ranking individuals show higher rates, upward arrows indicate that higher-ranking individuals show higher rates. Interactions can show Down The Hierarchy (DTH; Targeting lower-ranking individuals), Closely Ranked Receiver (CRR; Targeting individuals with similar rank); Up The Hierarchy (UTH; Targeting higher-ranking individuals), or a mix of those.

| Model | Standardization | DeltaLOOs | Sender Rank | Receiver Rank | Interaction |
|---|---|---|---|---|---|
| *Factor Higher-ranking* | - | 180.0 | ↓ | - | DTH |
| *Main Effects* | Raw | 183.9 | ↑ | ↑ | - |
| | Ordinal | 100.3 | ↓ | ↓ | - |
| | Proportional | 105.2 | ↓ | ↓ | - |
| *Rank Difference* | Raw | 176.8 | ↑ | - | DTH |
| | Ordinal | 97.2 | ↓ | - | UTH |
| | Proportional | 66.5 | ↓ | - | UTH |
| *Absolute Rank Difference* | Raw | 207.9 | - | - | - |
| | Ordinal | 197.3 | ↓ | - | CRR |
| | Proportional | 197.2 | ↓ | - | CRR |
| *Interaction* | Raw | 185.3 | ↑ | ↑ | CRR |
| | Ordinal | 123.3 | ↓ | ↓ | CRR for low-ranking |
| | Proportional | 111.7 | ↓ | ↓ | CRR for low-ranking |
| *Nonlinear Interaction* | Raw | 181.0 | - | - | CRR |
| | Ordinal | 33.6 | ↓ | ↓ | CRR for low-ranking |
| | Proportional | 0 | ↓ | ↓ | CRR for low-ranking |

about a species and social evolution more broadly [18]. While I make some of the choices in this study explicit and show their impact, other researcher choices remained fixed here but certainly influenced results (such as the exclusion of certain age/sex classes; [49].

Studies of animal sociality and evolution are, at their core, comparative: we study one species with the goal of understanding underlying evolutionary processes across species. For this, it is vital that results across studies are comparable [16, 17]. A positive note of this study is that, while Elo ratings and David's scores differed in predictive power (assessed using the out-of-sample predictive performance of each model using leave-one-out cross validation information criterion, and the within-sample explained variance), they would have led to a similar interpretation of results. The same was true for the use of proportional and ordinal dominance ranks (while using raw values would have led to different results). Thus, at least for systems with clearly linear hierarchy and sufficient data, these choices might only weakly influence interpretation. They might however still be relevant in cases where one of them leads to significant results in a frequentist framework, while the others do not, and researchers select reported models based on this hidden multiple comparison [20]. More worrisome, I found that differences in model specification caused considerable differences in results and interpretation. Most model specifications can only represent one type of relationship. For example, a researcher using the absolute rank difference would find that all interaction types in sooty mangabeys are directed at closely ranked group members. A researcher using the simple rank difference would find that aggression and supplants go down the hierarchy, while grooming goes up the hierarchy, and would not be able to make meaningful statements about proximity at all. Most likely, these effects are both present in most systems, but we would class the same social system differently based on these results.

In my comparisons, no combination of index/standardization/model specification outperformed the others throughout–however, some choices consistently performed worse than others did. Raw Elo ratings and David's Scores performed poorly, indicating that the distance between individual values generated by those scores was not meaningful, with equidistance the

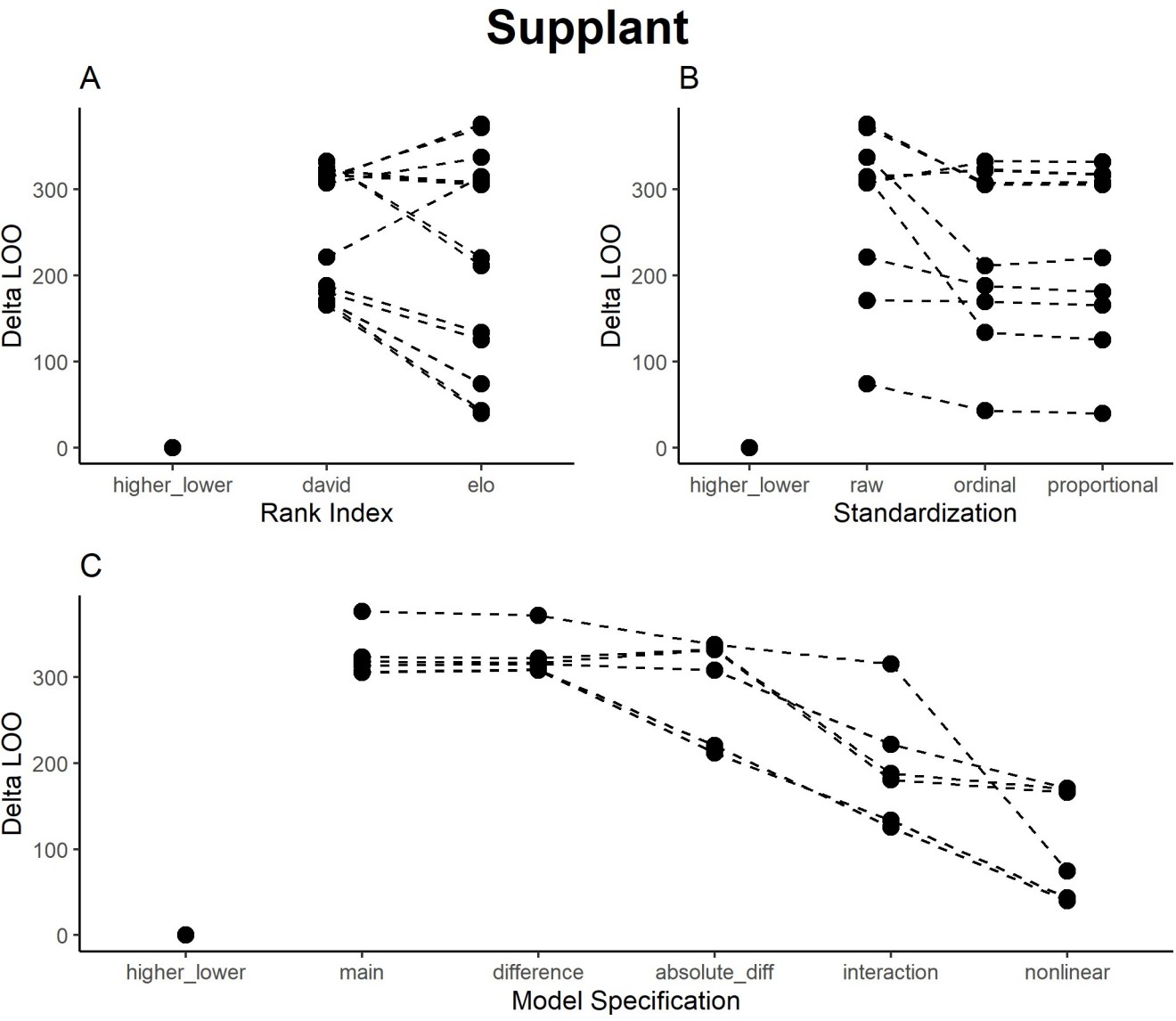

**Fig 4. Results of model comparison for supplant interactions.** The y-axis portrays delta LOO-ICs (distance to best model)—the best model is therefore set to 0, and higher scores indicate poorer performance. Points indicate models (split by the respective category), while lines connect otherwise comparable models (e.g., the main effects ordinal scale models for both Elo ratings and David's Scores).

most parsimonious assumption. There is no indication that the numeric distances produced by some human-made algorithms are meaningful for how animals structure their dominance hierarchies, and raw values will be influenced to a larger degree by measurement error, so equidistance is the assumption that minimises error in this case. It is worth remembering that raw Elo ratings and David's Scores are relative measures of winning likelihoods, which implies that the lower-ranking individual could, in principle, win the contest. While this may be true for aggression in several animal species, it is not true for many behaviours that are used to create dominance hierarchies in primates, which are strictly one-directional, transitive and are acknowledgements of the lower-ranking individual of the higher-ranking individual's position (such as displacements/supplants or pant grunts; [50]). This has implications on common indices that use raw David's Scores to test hierarchy steepness [36] and assume that the

**Table 5. Interpretation of Elo results for supplants.** Model interpretations are displayed for all models using Elo ratings (similar results for David's Scores). Downward arrows indicate that lower-ranking individuals show higher rates, upward arrows indicate that higher-ranking individuals show higher rates. Interactions can show Down The Hierarchy (DTH; Targeting lower-ranking individuals), Closely Ranked Receiver (CRR; Targeting individuals with similar rank); Up The Hierarchy (UTH; Targeting higher-ranking individuals), or a mix of those.

| Model | Standardization | DeltaLOO | Sender Rank | Receiver Rank | Interaction |
|---|---|---|---|---|---|
| *Factor Higher-ranking* | - | 0 | ↑ | - | DTH |
| *Main Effects* | Raw | 376.2 | ↑ | ↓ | - |
| | Ordinal | 305.3 | ↑ | ↓ | - |
| | Proportional | 305.6 | ↑ | ↓ | - |
| *Rank Difference* | Raw | 371.9 | - | - | DTH |
| | Ordinal | 307.9 | - | - | DTH |
| | Proportional | 308.5 | - | - | DTH |
| *Absolute Rank Difference* | Raw | 337.7 | ↑ | - | High-ranking: CRR and DTH |
| | Ordinal | 211.4 | ↑ | - | High-ranking: CRR and DTH |
| | Proportional | 220.5 | ↑ | - | High-ranking: CRR and DTH |
| *Interaction* | Raw | 315.0 | ↑ | ↓ | High-ranking: CRR and DTH; Low-ranking: CRR |
| | Ordinal | 133.7 | ↑ | ↓ | High-ranking: CRR and DTH; Low-ranking: CRR |
| | Proportional | 125.5 | ↑ | ↓ | High-ranking: CRR and DTH; Low-ranking: CRR |
| *Nonlinear Interaction* | Raw | 74.3 | ↑ | ↓ | CRR and DTH |
| | Ordinal | 43.0 | ↑ | ↓ | CRR and DTH |
| | Proportional | 39.8 | ↑ | ↓ | CRR and DTH |

differences between individuals are meaningful. However, it is worth investigating whether the effect presented here holds across species and whether it would be replicated when using contest competition to create hierarchies.

Like the raw values, main effects and simple rank distance models performed poorly throughout–often, these choices had poor predictive accuracy and failed to detect the most likely pattern (based on models with higher predictive accuracy). The absolute rank distance and higher/lower factor models performed well for a subset of the models but are limited in the information they could encode–they will invariably lead to the same, simple interpretation, no matter the underlying data. The higher/lower factor even 'found' a directional pattern in symmetrical proximity data. The interaction and nonlinear terms were able to detect relatively complex patterns and performed reasonably well for all interaction types, so if researchers would want to apply one model specification without previous knowledge of which patterns to expect, nonlinear terms would be the best choice. This is largely due to nonlinear patterns in interaction distributions, probably arising out of the interplay of competition and kinship patterns [51]. However, it is hard to interpret these more complex models unambiguously, and there is a risk of overfitting and poor prediction out-of-sample.

Neither Elo ratings nor David's Scores performed better across the board, and the same was true for ordinal or proportional values. Often, within the same comparison, some ordinal models performed better than the proportional models while some performed worse, with no clear pattern explaining these differences (especially given that the two standardizations correlated more than -0.99). This highlights the danger of using model comparisons to identify the 'best predictor' for any given distribution [13]: once we complexify the picture, it soon becomes hard to find meaningful explanations for the observed patterns, and any additional choice could upend the interpretation. Recently, studies have interpreted the difference in model performance using different rank variables or standardizations as a sign that the power structures or individual decisions within the group make the same assumptions as that index [13, 28]. However, these conclusions have to be drawn with care: comparisons between models

indicate which model has lower error in-sample, but they do not provide evidence that either model is particularly good at representing the social structure or decision-making of a group [52]. Higher predictive accuracy does not allow us to assume causation [53]. For example, most indices we currently use assume linear hierarchies [7], which is often not assessed [28]. We do not, currently, know how much sampling and measurement errors bias dominance indices and their performance. Most indices and standardizations are correlated very highly [11], especially when sufficient data are available and the linearity assumption is fulfilled, so any difference in their performance could be the result of random variation of few data points. In my models, the higher/lower factor often performed best, but would entirely fail to predict aggression or supplant patterns within the subset of dyads where the sender was higher-ranking.

For the mangabeys, these results paint a complex picture of the impact of rank on social interactions. The results are broadly in line with earlier studies for this species [31–35, 54]. My previous use of absolute rank differences to describe preferred association patterns [32] might have omitted important information about differences in social behaviour between individuals of different ranks. In this study, the effect we previously found for female-female association patterns (preferred spatial association with closely ranked group members) only held for lower-ranking group members–possibly because high-ranking females associate more closely with males, which were omitted here [32]. Social interactions (both socio-positive and nega-tive) occurred largely with closely ranked group members. Aggression and supplants were directed down the hierarchy by high-ranking group members, while low-ranking individuals groomed up the hierarchy. The grooming results (strong preference for close rank, more pro-nounced in high-ranking individuals) are a replication of earlier results using a different ana-lytical approach [31] and are in line with some predictions for cercopithecine monkeys with similar social systems [51]. However, models including other factors would be necessary to determine whether the preference for closely ranked individuals is the result of genuine prefer-ence for closely ranked partners, attraction to kin, reciprocity, spatial proximity, or priority of access.

Given that many of the steps taken here are rarely described in detail in ecological studies, and data and scripts are still mostly unavailable in this field [55], we face further challenges to replicability. One way to counter these problems to replicability would be to report all possible dominance rank index/standardization/model specification combinations in some form of 'multiverse' analysis [14, 24]. Given that analyses these days are done using some statistical software, repeating the analyses with all possible combinations does not in itself pose a compu-tational problem (even though it might dramatically increase computation times). This approach has the advantage of increasing transparency and making results comparable across studies because researchers can compare the same model specifications against each other, rather than different specifications. The same is true for any analysis that includes dominance rank, or any other index calculated by researchers (e.g., Dyadic Sociality Indices). Interpreta-tion can become more difficult and ambiguous, as I have shown in this study–at the same time, researchers can demonstrate that the interpretation they are presenting is not conditional on the choices they made in the data pipeline [18]. Further developing this framework will be an ongoing process to improve research in ecology and evolutionary biology [14, 18].

## Supporting information

**S1 Checklist. Inclusivity in global research.**
(DOCX)

**S1 File.**
(DOCX)

## Acknowledgments

The author thanks the Ivorian Ministry of Environment and Forests and Ministry of Higher Education and Scientific Research and the Office Ivoirien des Parcs et Reserves of Côte d'Ivoire. I thank Simon Kannieu, Daniel Bouin, Gnimion Florent, and the team of the TCP for field work support and data collection. Many thanks go to Roman M Wittig and Catherine Crockford for making this research possible and comments on the manuscript, and to Delphine de Moor for helpful comments.

## Author Contributions

**Conceptualization:** Alexander Mielke.

**Data curation:** Alexander Mielke.

**Formal analysis:** Alexander Mielke.

**Funding acquisition:** Alexander Mielke.

**Investigation:** Alexander Mielke.

**Methodology:** Alexander Mielke.

**Project administration:** Alexander Mielke.

**Resources:** Alexander Mielke.

**Software:** Alexander Mielke.

**Visualization:** Alexander Mielke.

**Writing – original draft:** Alexander Mielke.

**Writing – review & editing:** Alexander Mielke.

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
