## [Decision Letter · Decision Letter 0]

8 Jun 2022

PONE-D-22-07922Impact of dominance rank specification in dyadic interaction modelsPLOS ONE

Dear Dr. Mielke,

Thank you for submitting your manuscript to PLOS ONE. After careful consideration, we feel that it has merit but does not fully meet PLOS ONE’s publication criteria as it currently stands. Therefore, we invite you to submit a revised version of the manuscript that addresses the points raised during the review process. As you can see both reviewers and favourably impressed by your manuscript. However, they ask for more clarity about the rationale at the basis of your choice. R1 suggests presenting clearer practical information on when using the different methods and making them easier to a broad range of scholars. R2 strongly suggests discussing the empirical data into a more general framework by reviewing the literature available on the topic. I think that the reviewers' requests are not difficult to address. 

Please submit your revised manuscript by Jul 23 2022 11:59PM If you will need more time than this to complete your revisions, please reply to this message or contact the journal office at plosone@plos.org. Please include the following items when submitting your revised manuscript:A rebuttal letter that responds to each point raised by the academic editor and reviewer(s). You should upload this letter as a separate file labeled 'Response to Reviewers'.A marked-up copy of your manuscript that highlights changes made to the original version. You should upload this as a separate file labeled 'Revised Manuscript with Track Changes'.An unmarked version of your revised paper without tracked changes. You should upload this as a separate file labeled 'Manuscript'.

We look forward to receiving your revised manuscript.

Kind regards,

Elisabetta Palagi

Academic Editor

PLOS ONE

Reviewers' comments:

Reviewer's Responses to Questions

**Comments to the Author**

1. Is the manuscript technically sound, and do the data support the conclusions?

Reviewer #1: Yes

Reviewer #2: Partly

2. Has the statistical analysis been performed appropriately and rigorously? 

Reviewer #1: Yes

Reviewer #2: I Don't Know

3. Have the authors made all data underlying the findings in their manuscript fully available?

Reviewer #1: Yes

Reviewer #2: Yes

4. Is the manuscript presented in an intelligible fashion and written in standard English?

Reviewer #1: Yes

Reviewer #2: Yes

5. Review Comments to the Author

Reviewer #1: ABSTRACT

INTRODUCTION

LINE 48: Spending few more words on this concept could be useful to the reader, maybe also mentioning a research example for the issue

LINE 57-58: In which way could a pre-analytical assumption introduce a bias in the results obtained on the social system? Do you refer, for instance, to cases when you apply a specific tool/method to obtain NDS values or indices for the linearity of hierarchies even when the system is not suitable for applying them? Please specify better.

LINE 73: I think ‘of’ is a typo here.

LINE 110: when you refer to ‘different model specifications of rank differences’ are you referring to the three options ‘raw index values, proportional ranks, ordinal ranks’? If so, I suggest specifying it and to anticipate here what the three options properly mean and what they differ from each other.

LINE 123: please specify how (in terms of statistical tool / h or transitivity index or others) the authors found female hierarchies to be ‘clearly linear’, since, as you discuss in the current manuscript, results are susceptible to changes according to the statistical methods used and assumptions made.

METHODS

LINE 195: I think it could be useful for the reader to have a clearer description of what the response variable consisted of. For instance, was the ‘aggression events’ response variable a numerical variable for all the possible dyads of your n female individuals (with a-b dyad different from b-a) with the number of aggressive interactions done from a to b (or vice versa) for each dyad?

LINE 210: Why you chose to use as random effects sender identity and receiver identity separated and not as the interaction between these two?

RESULTS/DISCUSSION

What do you think could be the theoretical reasoning for proportional and ordinal values to be better than the raw ones, since the first two require the assumption of equidistance between ranks? I may think this results is appliable only to highly linear as well as highly steep hierarchies in which the distance between the ranks are discrete and evident?

The strong determinant of the study is the way you assessed the ‘quality’ of the different models (as you discuss there are no universal objective ways of predicting model quality), but how can you say the one you used is at least the best one)? Please, if possible, contextualize your results in the framework of the method used to discriminate ‘better’ and ‘worse’ models.

As you introduce at LINE 30-32 (Introduction), hierarchical ranks (obtained as NDS or Elo ratings) are highly used in ethology to, for instance, study how the distribution of behaviours is influenced by the position of the individuals along the hierarchy (e.g., when using GLMMs the NDS is often inserted as a fixed effect). What I do not precisely get from your work is whether and how your conclusions on the suitability of using NDS vs Elo rating or raw vs proportional vs ordinal indices could also apply to studies that use indices of hierarchical rank (i.e., NDS) to analyse the distribution of some behaviours. Are you suggesting Elo ratings over NDS and ordinal/proportional values over raw ones to be used in general in models (e.g., when using hierarchical ranks as fixed effect in GLMMs) or only when applying models set such as those you used to study the different interactions (e.g., aggressive) in the present study? I think this should be important to be specified in order to broaden even more the usefulness of this work.

Reviewer #2: In this study, the author tackles an important, methodological aspect of behavioral ecology related to quantitative approaches to evaluate and use dominance ranks and ranking-differences among individuals in a group in statistical analyses. The work no doubt has its merits, and the questions asked are novel in spite of the now broad, ever growing literature on methodological considerations while evaluating hierarchies. Beyond just comparing ranking methods as extensive studies have now done, the author (somewhat exploratorily, in my opinion) attempts to evaluate how researchers’ choices pertaining to evaluating measures of dominance might impact the performance of linear mixed effects models when one or the other of these measures are used. In doing so, the goal is to highlight discrepancies in dominance-related findings across studies that arise merely as a consequence of pre-processing decisions of how best to evaluate ranks.

While I agree that such an effort would doubtless be useful to animal behavioral studies, I nevertheless have some concerns regarding the authors decisions, choices, and rationale as explained in the Introduction and Methods sections. I have summarized these concerns below. Should the author address these in a revised manuscript, I would be happy to review the revisions to these sections as well as provide a more in-depth review of the Results and Discussion section at the time.

First, I agree in principle that researchers have made different choices – e.g. absolute ranks, rank index, standardized rank-differences etc. – across studies. That said, I am left wondering about the extent to which differences in these choices across studies stem more so from hypotheses-driven (rather than random, as seems to be the author’s primary argument here) decision-making. For instance, one might think that researchers would be more prone to using percentile (rather than absolute or ordinal) ranks in a comparative study involving multiple groups and species, as opposed to a study that's conducted in a one-off group for example. To validate this or not, perhaps a more interesting endeavor that I think should probably precede these analyses would be to write a review article. This could potentially focus on studies (say over the past 5-10 years) on nonhuman primates, but also other group-living animals in which dominance hierarchies have been evaluated, and summarize pertinent information from these studies – e.g. (i) computed dominance ranks, (ii) the method they have used, (iii) the rank-related measures they have used in their data analyses (GLMMs, correlations etc.), and most importantly (iv) the rationale they offer for these decisions, if any. This would establish a clearer premise. In terms of how often these choices are random versus question- or hypothesis-driven, for conducting this analysis.

Without such a review of the literature, as a researcher I would have a limited understanding of the depth of the problem that the author tackles empirically here. Without a review, I’d give credit to researchers insofar as simply expecting each of them to think about their question, hypotheses and predictions, and come up a suitable "simplest" level of analysis to address these hypothesis (e.g. individual, dyad), and finally the measures that would be most well suited given the above.

Moreover, I am somewhat concerned that this study has been restricted to just female-female interactions for a single, Cercopithecine primate taxon in which females are expected to show linear (if not steep) dominance hierarchies. I think evaluating robustness (or lack thereof) to hierarchical variation are key to “drive home” methodological assessments of dominance, but such variation is clearly not accounted for by the reductionist or restrictive approach taken here. Most methodological discrepancies in estimations of dominance (whether ranking methods, or the use of different quantitative aspects of dominance success as has been done here) arise from noninteracting dyads, changes in group composition, or among animals that interact consistently but infrequently. Given this, wouldn’t the inclusion of males, at the very least, be necessary to validate your findings?

Third, I see that you have only conducted these assessments for dyadic differences. This in itself would entail, to the best of my knowledge, using matrix-regression approaches in the place of, or at least in addition to, your approach of using linear regression in which you include the attribute (rank) of one animal and various computations of differences in rank between that animal (giver/sender) and the other member of the dyad. Why not use matrix regressions, as several previous studies have done? Explain more clearly as to why your approach would be better than, say, a matrix-based approach like Mantel tests and/or an MR-QAP regression. A second, closely related point - in addition to, or instead of, dyadic rank-differences, why not conduct similar assessments for absolute ranks at the individual level, to minimize dyadic inter-dependency issues? For instance, as you say earlier, compare results of analyses in which you include ordinal ranks, rank indices (ranging between 0 -> 1), and cardinal scores (e.g. DS), to examine the effects of each of these on individual animals' (i) aggression given, (ii) aggression received, (iii) grooming given, (iv) grooming received etc.?

Finally, a potentially minor point. I see that you included observation effort as an offset term in your models. I take it that observation effort was not equal across both members of dyads, so did you then use the value for just one member of the dyad? This seems a bit unclear.

6. PLOS authors have the option to publish the peer review history of their article (what does this mean?). If published, this will include your full peer review and any attached files.

Reviewer #1: **Yes: **Luca Pedruzzi

Reviewer #2: No

---

## [Author Response · Author response to Decision Letter 0]

19 Jul 2022

Reviewers' comments:

Reviewer's Responses to Questions

Comments to the Author

Reviewer #1: ABSTRACT

INTRODUCTION

LINE 48: Spending few more words on this concept could be useful to the reader, maybe also mentioning a research example for the issue

- I have now added an example for these effects (L51): ‘For example, in tufted capuchin females, high-ranking individuals gain more grooming (effect of dominance rank), individuals groom similarly-ranked individuals more than expected (effect of rank distance), but high- and low-ranking individuals differ in how freely they can choose partners (non-linear effect of groomer rank; Tiddi et al., 2012).’

LINE 57-58: In which way could a pre-analytical assumption introduce a bias in the results obtained on the social system? Do you refer, for instance, to cases when you apply a specific tool/method to obtain NDS values or indices for the linearity of hierarchies even when the system is not suitable for applying them? Please specify better.

- I have now added further explanations based on the above example (L61): ‘For example, for the above example of tufted capuchin females (Tiddi et al., 2012), if a researcher models only the dominance ranks of the two groomers, they might find that high-ranking individuals received more grooming. With the same data, another researcher who included only the rank difference would interpret the results to indicate that individuals were attracted to similarly-ranked group members. In isolation, these would point to fundamentally different processes underlying grooming partner choice.’

LINE 73: I think ‘of’ is a typo here.

- Thank you, removed.

LINE 110: when you refer to ‘different model specifications of rank differences’ are you referring to the three options ‘raw index values, proportional ranks, ordinal ranks’? If so, I suggest specifying it and to anticipate here what the three options properly mean and what they differ from each other.

- I have now clarified this in more detail (L112ff): ‘I compared different standardizations - raw David’s Scores or Elo rating values, proportional ranks (all ranks equidistant between 0 and 1, with 1 for the highest-ranked individual), and ordinal ranks (1 for highest-ranking individual, 2 for second highest etc) - because this choice has recently been shown to influence model outcomes in primate studies (Levy et al., 2020; Schino & Lasio, 2019), but insufficient information still exists on whether and how they influence models that represent interaction rates between dyads of individuals within groups. My focus lies on how the power relations between the individuals in a dyad are represented when fitting models (whether as main effects, rank distance variables, or interaction terms; see Table 1), given that no systematic information on the impact of this choice is currently available.’

LINE 123: please specify how (in terms of statistical tool / h or transitivity index or others) the authors found female hierarchies to be ‘clearly linear’, since, as you discuss in the current manuscript, results are susceptible to changes according to the statistical methods used and assumptions made.

- I have now specified this (L132): ‘I only use adult females, as their hierarchies have been found to be clearly linear in a neighbouring community (h = 0.71; Range & Noë, 2002) and this community (only 0.6% of supplants direct against the hierarchy; Mielke, 2022 unpublished data) and they interact at relatively high rates, while the connection between male and female hierarchies is less clear and males interact at low rates in this group (Mielke et al., 2017).’

METHODS

LINE 195: I think it could be useful for the reader to have a clearer description of what the response variable consisted of. For instance, was the ‘aggression events’ response variable a numerical variable for all the possible dyads of your n female individuals (with a-b dyad different from b-a) with the number of aggressive interactions done from a to b (or vice versa) for each dyad?

- I have added a paragraph (L131ff) clarifying that the response variables here are numeric counts of events/minutes of A interacting with B in any given year. 

LINE 210: Why you chose to use as random effects sender identity and receiver identity separated and not as the interaction between these two?

- I have added dyad identity as random effect and updated all modelsnnd. 

RESULTS/DISCUSSION

What do you think could be the theoretical reasoning for proportional and ordinal values to be better than the raw ones, since the first two require the assumption of equidistance between ranks? I may think this results is appliable only to highly linear as well as highly steep hierarchies in which the distance between the ranks are discrete and evident?

- I think this might be a consistent finding, even in non-linear hierarchies. The reason is probably that our rank indices have quite a huge measurement error attached to them – the actual Elo and DS values are sample size dependent and probably undergo variation with every missed data point. There is no biological reason to assume that an individual with an Elo rating of 600 is half as powerful as one with an Elo rating of 1200, that a 50 point jump at any point of the scale is that same, or that primates perceive power on that scale. Also, the point distance between the first and fifth individual is influenced by the interactions and sample size of interactions of individuals between them, which might be biologically meaningless. Thus, while establishing a reliable rank order can probably be done with the sample sizes we usually observe, estimating the exactly correct distance between two individuals might be much more data intensive and involve very different assumptions. I have added a sentence in the Discussion (L416): ‘There is no indication that the numeric distances produce by some human-made algorithms are meaningful for how animals structure their dominance hierarchies, and raw values will be influenced to a larger degree by measurement error, so equidistance is the assumption that minimises error in this case.’

The strong determinant of the study is the way you assessed the ‘quality’ of the different models (as you discuss there are no universal objective ways of predicting model quality), but how can you say the one you used is at least the best one)? Please, if possible, contextualize your results in the framework of the method used to discriminate ‘better’ and ‘worse’ models.

- I have added information about the used performance criteria for the models (out-of-sample prediction accuracy and within-sample explained variance) in both the methods (L242: ‘My assumption here is that, given that the outcome remains constant, the model with the smallest LOO-IC predicts the outcome most accurately, and that models with lower leave-one out cross validation information criterion and higher Bayesian R2 (Gelman et al., 2019) can be considered to be ‘better’ than models with higher LOO-IC or lower Bayesian R2.’) and the discussion (L401: ‘A positive note of this study is that, while Elo ratings and David’s scores differed in predictive power (assessed using the out-of-sample predictive performance of each model using leave-one-out cross validation information criterion, and the within-sample explained variance)…’)

As you introduce at LINE 30-32 (Introduction), hierarchical ranks (obtained as NDS or Elo ratings) are highly used in ethology to, for instance, study how the distribution of behaviours is influenced by the position of the individuals along the hierarchy (e.g., when using GLMMs the NDS is often inserted as a fixed effect). What I do not precisely get from your work is whether and how your conclusions on the suitability of using NDS vs Elo rating or raw vs proportional vs ordinal indices could also apply to studies that use indices of hierarchical rank (i.e., NDS) to analyse the distribution of some behaviours. Are you suggesting Elo ratings over NDS and ordinal/proportional values over raw ones to be used in general in models (e.g., when using hierarchical ranks as fixed effect in GLMMs) or only when applying models set such as those you used to study the different interactions (e.g., aggressive) in the present study? I think this should be important to be specified in order to broaden even more the usefulness of this work.

- As I say in the concluding paragraph of the manuscript (L474 onwards), I think the best approach would be to use a multiverse approach where researchers do not decide to use either NDS or Elo, for example, or just one way of standardising the indices, but use and report all the different options. I have no reason to believe that either NDS or Elo is ‘better’ across all possible studies (barring cases where ranks can be modelled dynamically), and that is one thing my results clearly show. However, we face a replicability problem if researcher calculate both and just report the one that supported their predictions. This will be the same for any kind of analysis that includes rank. I have added some information to clarify this: ‘One way to counter these problems to replicability would be to report all possible dominance rank index/standardization/model specification combinations in some form of ‘multiverse’ analysis (Hoffmann et al., 2021; Steegen et al., 2016). Given that analyses these days are done using some statistical software, repeating the analyses with all possible combinations does not in itself pose a computational problem (even though it might dramatically increase computation times). This approach has the advantage of increasing transparency and making results comparable across studies because researchers can compare the same model specifications against each other, rather than different specifications. The same is true for any analysis that includes dominance rank, or any other index calculated by researchers (e.g., Dyadic Sociality Indices).’

Reviewer #2: In this study, the author tackles an important, methodological aspect of behavioral ecology related to quantitative approaches to evaluate and use dominance ranks and ranking-differences among individuals in a group in statistical analyses. The work no doubt has its merits, and the questions asked are novel in spite of the now broad, ever growing literature on methodological considerations while evaluating hierarchies. Beyond just comparing ranking methods as extensive studies have now done, the author (somewhat exploratorily, in my opinion) attempts to evaluate how researchers’ choices pertaining to evaluating measures of dominance might impact the performance of linear mixed effects models when one or the other of these measures are used. In doing so, the goal is to highlight discrepancies in dominance-related findings across studies that arise merely as a consequence of pre-processing decisions of how best to evaluate ranks.

While I agree that such an effort would doubtless be useful to animal behavioral studies, I nevertheless have some concerns regarding the authors decisions, choices, and rationale as explained in the Introduction and Methods sections. I have summarized these concerns below. Should the author address these in a revised manuscript, I would be happy to review the revisions to these sections as well as provide a more in-depth review of the Results and Discussion section at the time.

First, I agree in principle that researchers have made different choices – e.g. absolute ranks, rank index, standardized rank-differences etc. – across studies. That said, I am left wondering about the extent to which differences in these choices across studies stem more so from hypotheses-driven (rather than random, as seems to be the author’s primary argument here) decision-making. For instance, one might think that researchers would be more prone to using percentile (rather than absolute or ordinal) ranks in a comparative study involving multiple groups and species, as opposed to a study that's conducted in a one-off group for example. To validate this or not, perhaps a more interesting endeavor that I think should probably precede these analyses would be to write a review article. This could potentially focus on studies (say over the past 5-10 years) on nonhuman primates, but also other group-living animals in which dominance hierarchies have been evaluated, and summarize pertinent information from these studies – e.g. (i) computed dominance ranks, (ii) the method they have used, (iii) the rank-related measures they have used in their data analyses (GLMMs, correlations etc.), and most importantly (iv) the rationale they offer for these decisions, if any. This would establish a clearer premise. In terms of how often these choices are random versus question- or hypothesis-driven, for conducting this analysis.

- While I thank the reviewer for their opinion, I do not consider ‘write a different paper’ feasible advice for peer review. This is a methods paper that will be of use for researchers who are wondering how to best specify rank differences in their statistical models – it should be judged based on whether it will achieve this or not. I assume that these researchers will have an understanding of the literature in their field. That aside, I address the deeper point here (that there might be hypothesis-driven reasons to choose one model over the other) throughout the manuscript – if researchers build their predictions into the model, we are biasing the interpretations of our work. If I only use absolute rank difference as a model parameter (because that is my hypothesis), I will always find that similarly-ranked individuals interact more – by having selected that model parameter over others, I have biased the possible interpretations I can arrive at before even seeing the data. Researchers need to be aware of this bias, no matter what their hypotheses are.

Without such a review of the literature, as a researcher I would have a limited understanding of the depth of the problem that the author tackles empirically here. Without a review, I’d give credit to researchers insofar as simply expecting each of them to think about their question, hypotheses and predictions, and come up a suitable "simplest" level of analysis to address these hypothesis (e.g. individual, dyad), and finally the measures that would be most well suited given the above.

- The function of this paper is to make researchers aware of the depth of the problem. My results clearly show that, if my analyses of mangabey social interactions had been done with one approach rather than another, they would have resulted in different interpretations of the data. This is worrisome as it limits our ability to interpret past results. There is ample research showing that questionable research practices are widespread in ecological research, as they are in many other research field, especially given that data and scripts are still usually not reported (see O’Dea, R. E., Parker, T. H., Chee, Y. E., Culina, A., Drobniak, S. M., Duncan, D. H., Fidler, F., Gould, E., Ihle, M., Kelly, C. D., Lagisz, M., Roche, D. G., Sánchez-Tójar, A., Wilkinson, D. P., Wintle, B. C., & Nakagawa, S. (2021). Towards open, reliable, and transparent ecology and evolutionary biology. BMC Biology, 19(1), 68). Acknowledging that this happens does not pre-suppose a lack of ability, knowledge, or good intentions from researchers, but is simply the result of a skewed publication process and academic traditions in different fields. Exploring the impact these choices have on results is a first step in making researchers aware that their choices influence their conclusions, so they can rectify them.

Moreover, I am somewhat concerned that this study has been restricted to just female-female interactions for a single, Cercopithecine primate taxon in which females are expected to show linear (if not steep) dominance hierarchies. I think evaluating robustness (or lack thereof) to hierarchical variation are key to “drive home” methodological assessments of dominance, but such variation is clearly not accounted for by the reductionist or restrictive approach taken here. Most methodological discrepancies in estimations of dominance (whether ranking methods, or the use of different quantitative aspects of dominance success as has been done here) arise from noninteracting dyads, changes in group composition, or among animals that interact consistently but infrequently. Given this, wouldn’t the inclusion of males, at the very least, be necessary to validate your findings?

- The goal of this paper is not to make definitive statements about mangabey social systems, but to illustrate that problems arise in the statistical modelling process. Using the female mangabeys as a toy example here serves a didactic purpose – I could have used simulated data, but I feel that the impact of the manuscript would be lessened. Almost all research papers in cercopithecines (and most primates, at that) will address interaction distributions within the ‘more social’ sex. Obviously, this practice can be debated, but single-sex single-group studies currently make up easily 90% of this research literature. If I want to direct the attention of the reader to the problem at hand here (model specifications influence interpretations), it makes sense to rely on one well studied system and show that even within this comparatively simple system, the way statistical models are built influences interpretation. In more complex systems or when including multiple groups or sexes, the problem is more acute. 

Third, I see that you have only conducted these assessments for dyadic differences. This in itself would entail, to the best of my knowledge, using matrix-regression approaches in the place of, or at least in addition to, your approach of using linear regression in which you include the attribute (rank) of one animal and various computations of differences in rank between that animal (giver/sender) and the other member of the dyad. Why not use matrix regressions, as several previous studies have done? Explain more clearly as to why your approach would be better than, say, a matrix-based approach like Mantel tests and/or an MR-QAP regression. A second, closely related point - in addition to, or instead of, dyadic rank-differences, why not conduct similar assessments for absolute ranks at the individual level, to minimize dyadic inter-dependency issues? For instance, as you say earlier, compare results of analyses in which you include ordinal ranks, rank indices (ranging between 0 -> 1), and cardinal scores (e.g. DS), to examine the effects of each of these on individual animals' (i) aggression given, (ii) aggression received, (iii) grooming given, (iv) grooming received etc.?

- I agree with the reviewer that conducting a similar analysis for individual ranks would be of interest – even though the impact of researcher choices in those models is much better understood (e.g., https://doi.org/10.1002/ajp.22160;
https://doi.org/10.1098/rspb.2020.1013) . But again, this paper is specifically designed for researchers who model dyadic interactions and should be evaluated as such. MR-QAP and other node-level permutation approaches have recently seen a flurry of criticism, as they do not allow for accurate effect size estimation and do not control for all kinds of non-independence (https://www.biorxiv.org/content/10.1101/2021.06.04.447124v2.abstract) . They are also mainly useful if the researcher is not interested in any further (especially categorical) variables, such as individual sexes and ages, which would usually be part of this kind of analysis.

Finally, a potentially minor point. I see that you included observation effort as an offset term in your models. I take it that observation effort was not equal across both members of dyads, so did you then use the value for just one member of the dyad? This seems a bit unclear.

- I have added information the observation effort is the dyadic interaction effort, as either individual could be the sender or receiver of interactions when either of the two individuals was followed (L227).

---

## [Decision Letter · Decision Letter 1]

22 Sep 2022

PONE-D-22-07922R1Impact of dominance rank specification in dyadic interaction modelsPLOS ONE

Dear Dr. Mielke,

Thank you for submitting your manuscript to PLOS ONE. After careful consideration, we feel that it has merit but does not fully meet PLOS ONE’s publication criteria as it currently stands. Therefore, we invite you to submit a revised version of the manuscript that addresses the points raised during the review process.

We look forward to receiving your revised manuscript.

Kind regards,

Elisabetta Palagi

Academic Editor

PLOS ONE

Journal Requirements:

Reviewers' comments:

Reviewer's Responses to Questions

**Comments to the Author**

1. If the authors have adequately addressed your comments raised in a previous round of review and you feel that this manuscript is now acceptable for publication, you may indicate that here to bypass the “Comments to the Author” section, enter your conflict of interest statement in the “Confidential to Editor” section, and submit your "Accept" recommendation.

Reviewer #1: All comments have been addressed

Reviewer #3: All comments have been addressed

2. Is the manuscript technically sound, and do the data support the conclusions?

Reviewer #1: Yes

Reviewer #3: Yes

3. Has the statistical analysis been performed appropriately and rigorously? 

Reviewer #1: Yes

Reviewer #3: Yes

4. Have the authors made all data underlying the findings in their manuscript fully available?

Reviewer #1: Yes

Reviewer #3: Yes

5. Is the manuscript presented in an intelligible fashion and written in standard English?

Reviewer #1: Yes

Reviewer #3: Yes

6. Review Comments to the Author

Reviewer #1: Thanks for clarifying the parts that were pointed out in the previous Review. The manuscript can now be useful for researchers dealing with social hierarchies in primates as well as other social mammals.

Reviewer #3: This paper investigates different dominance indices, dominance rank standardization and model specification by studying the effect of dyadic rank relationships on the distribution and patterns of interactions within a group. The aim of this paper is to show how different choices made by researchers (referred to as “researcher degrees of freedom”) can impact the interpretation of results even when analyses are based on the same data. To prove this point, this work uses data of behavioural interactions in sooty mangabeys, and specifically female-female interactions, excluding intersexual and male-male interactions. The author justifies the choice to restrict the analyses to this subset of data based on the species’ characteristics (linearity of female of female hierarchies and high rates of interactions among females).

The author fits multiple generalized linear mixed models within a Bayesian framework using different model specifications and variables. Then, he uses leave-one-out cross-validation information criterion (LOO-IC) and Bayesian R^2 to identify the model that explains the most variance in the data, which is then considered the “best” model. Thus, the closer each model performs to the “best” model (comparing LOO-IC and Bayesian R^2) the better that model is assumed to be in correctly identifying the processes present in the data. This allows to compare the validity of each result and its interpretation. However, models that performed close to, but were not the “best” model were still considered to provide insightful information on the dynamics of social interactions analysed if they highlighted alternative patterns that were not possible to be captured by simpler, “better” models. This can come across as a by-product of the fact that the outcomes of the analyses are contingent on the choices of analyses, for example you cannot find a pattern that your model is not built to capture. The author addresses this point in the discussion in lines 411-418. Thus, it is an integral part of the argument proposed by the manuscript, where researcher degrees of freedom impact interpretation of results. At its very core, I feel the argument is conceptual and it would be valid even if argued only theoretically, but the use of empirical data underlines it.

For each result obtained from different model specifications and rank standardizations the author carefully highlights the differences in interpretation and how some results could paint only a partial picture of the dynamics occurring in the data, which would have led to different conclusions on the social organization of the species. Thus, this work succeeds in showing the impact that researcher degrees of freedom have on the interpretation of results obtained from the same data, using an example study species. As a solution, the author proposes a ‘multiverse analyses’ approach for future studies, where researchers are suggested to conduct analyses using a different combination of pre-processing and modelling choices to validate their results outside specific methodological choices and to increase repeatability.

In conclusion, I think that this work is a valuable contribution to studies evaluating methodological validation within the field of animal behaviour. Secondarily, it provides information on the patterns of interactions that female sooty mangabeys have with other females in relation to their dominance rank, using the same ‘multiverse analyses’ approach suggested by the author. However, I mention this only as a secondary point as the inclusion of intersexual and male-male interactions could have resulted in more complete set of results in that sense, but I understand that was not the target aim of this manuscript.

I only have one minor comment:

It is shown that assuming equidistance among dominance ranks is a better predictor than calculating raw differences in the values of the dominance indices, and I find it very interesting. The author argues that “there is no indication that the numeric distances produce [sic] by some human-made algorithms are meaningful for how animals structure their dominance hierarchies, and raw values will be influenced to a larger degree by measurement error” (lines 423-424), which is argued to be supported by the results and discussed in great detail. However, the discussion would benefit from warranting some space to elaborate on the possible implications if such finding proved to be correct also for other species and systems, and thus potentially a general characteristic of social systems. For example, there are multiple cases in which such numerical distances are assumed to be meaningful in the literature, with implications on interpretation of results (which is a relevant topic for this paper). For example, the methodology proposed by de Vries and colleagues (2006) uses the difference in David’s score values among individuals as a basis to calculate hierarchical steepness, and it’s a widely used metrics with over 300 citations. In addition, this result may be dependent on the characteristics of this study species and of female-female interactions specifically (e.g., high linearity, possibly high steepness), although the author correctly restricts the statement to these analyses: “equidistance is the assumption that minimises error in this case,” lines 425-426. If this result were consistently replicated in multiple species with different characteristics, it would have interesting implications. I suggest the author expand the discussion on this point to allow also non specialist readers to understand the possible implications of this finding alongside future avenues of research. Also, as a small typo I think the author intended to use “produced” and not “produce” in the sentence (line 423).

References

de Vries, H., Stevens, J. M. G., & Vervaecke, H. (2006). Measuring and testing the steepness of dominance hierarchies. Animal Behaviour, 71(3), 585–592. https://doi.org/10.1016/j.anbehav.2005.05.015

7. PLOS authors have the option to publish the peer review history of their article (what does this mean?). If published, this will include your full peer review and any attached files.

Reviewer #1: **Yes: **Luca Pedruzzi

Reviewer #3: **Yes: **Tommaso Saccà

---

## [Author Response · Author response to Decision Letter 1]

6 Oct 2022

Reviewer #3

It is shown that assuming equidistance among dominance ranks is a better predictor than calculating raw differences in the values of the dominance indices, and I find it very interesting. The author argues that “there is no indication that the numeric distances produce [sic] by some human-made algorithms are meaningful for how animals structure their dominance hierarchies, and raw values will be influenced to a larger degree by measurement error” (lines 423-424), which is argued to be supported by the results and discussed in great detail. However, the discussion would benefit from warranting some space to elaborate on the possible implications if such finding proved to be correct also for other species and systems, and thus potentially a general characteristic of social systems. For example, there are multiple cases in which such numerical distances are assumed to be meaningful in the literature, with implications on interpretation of results (which is a relevant topic for this paper). For example, the methodology proposed by de Vries and colleagues (2006) uses the difference in David’s score values among individuals as a basis to calculate hierarchical steepness, and it’s a widely used metrics with over 300 citations. In addition, this result may be dependent on the characteristics of this study species and of female-female interactions specifically (e.g., high linearity, possibly high steepness), although the author correctly restricts the statement to these analyses: “equidistance is the assumption that minimises error in this case,” lines 425-426. If this result were consistently replicated in multiple species with different characteristics, it would have interesting implications. I suggest the author expand the discussion on this point to allow also non specialist readers to understand the possible implications of this finding alongside future avenues of research. Also, as a small typo I think the author intended to use “produced” and not “produce” in the sentence (line 423). 

- I thank the reviewer for this valuable observation! I have added this further information for readers to clarify this:

‘It is worth remembering that raw Elo ratings and David’s Scores are relative measures of winning likelihoods, which implies that the lower-ranking individual could, in principle, win the contest. While this may be true for aggression in several animal species, it is not true for many behaviours that are used to create dominance hierarchies in primates, which are strictly one-directional, transitive and are acknowledgements of the lower-ranking individual of the higher-ranking individual’s position (such as displacements/supplants or pant grunts; Mielke et al. 2019). This has implications on common indices that use raw David’s Scores to test hierarchy steepness (de Vries et al. 2006) and assume that the differences between individuals are meaningful. However, it is worth investigating whether the effect presented here holds across species and whether it would be replicated when using contest competition to create hierarchies.’ (L.423)

---

## [Editor Report · Decision Letter 2]

21 Oct 2022

Impact of dominance rank specification in dyadic interaction models

PONE-D-22-07922R2

Dear Dr. Mielke,

We’re pleased to inform you that your manuscript has been judged scientifically suitable for publication and will be formally accepted for publication once it meets all outstanding technical requirements.

Kind regards,

Elisabetta Palagi

Academic Editor

PLOS ONE

---

## [Editor Report · Acceptance letter]

31 Oct 2022

PONE-D-22-07922R2 

Impact of dominance rank specification in dyadic interaction models 

Dear Dr. Mielke:

I'm pleased to inform you that your manuscript has been deemed suitable for publication in PLOS ONE. Congratulations! Your manuscript is now with our production department. 

Kind regards, 

on behalf of

Dr. Elisabetta Palagi 

Academic Editor

PLOS ONE